# Understanding glioblastoma invasion using physically-guided neural networks with internal variables

**Jacobo Ayensa-Jiménez**[1,2,4], **Mohamed H. Doweidar**[1,2,5], **Jose A. Sanz-Herrera**[3], **Manuel Doblare**[1,2,4,5¤]*

**1** Mechanical Engineering Department, School of Engineering and Architecture, University of Zaragoza, Spain, **2** Aragón Institute of Engineering Research (I3A), University of Zaragoza, Spain, **3** Mechanical Engineering Department, School of Engineering, University of Sevilla, Spain, **4** Aragón Institute of Health Research (IIS Aragón), Spain, **5** Centro de Investigación Biomédica en Red en Bioingeniería, Biomateriales y Nanomedicina (CIBERBBN), Spain

¤ Current address: Mariano Esquillor, S/N, Zaragoza, Spain
* mdoblare@unizar.es

**Data Availability Statement:** All synthetic data used to train and validate the networks are available at Zenodo Data Repository (https://zenodo.org/

## Abstract

Microfluidic capacities for both recreating and monitoring cell cultures have opened the door to the use of Data Science and Machine Learning tools for understanding and simulating tumor evolution under controlled conditions. In this work, we show how these techniques could be applied to study Glioblastoma, the deadliest and most frequent primary brain tumor. In particular, we study Glioblastoma invasion using the recent concept of Physically-Guided Neural Networks with Internal Variables (PGNNIV), able to combine data obtained from microfluidic devices and some physical knowledge governing the tumor evolution. The physics is introduced in the network structure by means of a nonlinear advection-diffusion-reaction partial differential equation that models the Glioblastoma evolution. On the other hand, multilayer perceptrons combined with a nodal deconvolution technique are used for learning the *go or grow* metabolic behavior which characterises the Glioblastoma invasion. The PGNNIV is here trained using synthetic data obtained from *in silico* tests created under different oxygenation conditions, using a previously validated model. The unravelling capacity of PGNNIV enables discovering complex metabolic processes in a non-parametric way, thus giving explanatory capacity to the networks, and, as a consequence, surpassing the predictive power of any parametric approach and for any kind of stimulus. Besides, the possibility of working, for a particular tumor, with different boundary and initial conditions, permits the use of PGNNIV for defining virtual therapies and for drug design, thus making the first steps towards *in silico* personalised medicine.

## Author summary

In this work, we apply Physically-Guided Neural Networks with Internal Variables (PGNNIV) to the understanding of the Glioblastoma evolution process. We explain the metabolic changes between the proliferative and migrative activity of Glioblastoma cell

record/6349224#.Yi8NIbiCFhB). You can also use DOI:10.5281/zenodo.6349224.

**Funding:** MD and JASH gratefully acknowledge the financial support from the Spanish Ministry of Science and Innovation (MICINN) and FEDER, UE through the project PGC2018-097257-B-C31. MHD acknowledges the financial support from the Spanish Ministry of Science and Innovation (MICINN) and FEDER, through the project PID2019-106099RB-C44/AEI/10.13039/501100011033. All authors acknowledge the funding from the Government of Aragon and the Centro de Investigacion Biomedica en Red en Bioingenieria, Biomateriales y Nanomedicina (CIBER-BBN). CIBER-BBN is financed by the Instituto de Salud Carlos III with assistance from the European Regional Development Fund. The funders had no role in study design, data collection and analysis, decision to publish, or preparation of the manuscript.

**Competing interests:** The authors have declared that no competing interests exist.

cultures by using the *go or grow* activation functions as a pair of internal variables, whose dependence on the oxygen level is unravelled by some building blocks of the whole PGNNIV. Due to its model-free nature, our method is able to identify different classical mechanistic approaches and to outperform cell culture evolution predictions, as we demonstrate in the paper. Unlike Biologically-Informed Neural Networks we can assimilate data obtained from different boundary conditions and under different external stimuli to simulate the tumor progression under arbitrary conditions. We demonstrate this ability by comparing the predictions with different boundary conditions, resulting in different oxygenation conditions. This flexibility enables the use of our proposed method for personalised medical purposes, as the cell culture metabolic information, for a particular tumor, is encapsulated in a sub-network and may be used for arbitrary *in silico* tests.

This is a *PLOS Computational Biology* Methods paper.

## Introduction

Cancer is the second leading cause of death in the world, according to the World Health Organisation, and is responsible for about 10 million deaths per year. These figures are expected to rise up to 16 million deaths in 2040 [1]. Among the more than 200 types of cancer, Glioblastoma (GBM) is the most aggressive primary brain tumor, with a survival median of GBM patients who undergo the first-line standard treatment (surgery followed by adjuvant chemotherapy and local radiation) of 14 months since diagnosis, and a 5-year survival rate of only 6.8% [2, 3]. In addition, it is the most frequent of glioma tumors, accounting for 17% of this type of cancers [4]. Clinicians and researchers associate this high aggressiveness with its heterogeneity, rapid and dynamic progression and high invasive capacity [5, 6]. It is therefore clear that the characterization of GBM behavior is crucial for the development of therapeutic strategies against this cancer [7, 8].

The complexity of GBM evolution (and of cell biological processes in general), strongly depends on the particular microenvironment [9]. This makes it difficult the use of two-dimensional *in vitro* experiments (Petri dishes) to reproduce the actual behavior of cells in real tissues. In response to these limitations, microfluidics and micro-technologies permit to recreate the actual three-dimensional cell microenvironment in a much more realistic way, thus allowing to get results, associated with the complex biophysical and biochemical cell processes that govern the tumor dynamics, much closer to the actual *in vivo* situation than classical Petri dishes [10]. For instance, the study of tumor chemotaxis has been considerably developed [11, 12]. Consequently, and in general, these techniques allow testing drugs much more efficiently [13, 14].

Also, thanks to the flexibility, reproducibility, automation, integration and miniaturisation of microfluidic experiments, it is possible to generate great amounts of data, in the form of images and videos of cell culture evolution [15]. This opens the door to using Data Science and Machine Learning methods as new predicting tools [16, 17]. These tools can be considered therefore as a new paradigm in the analysis of complex, multifactorial, multiphysical and multiparametric biological problems, as those occurring during GBM evolution [18]. Therefore, it begins to be possible working in applied mathematical biology with complex nonlinear models, typical of other disciplines such as computational mechanics. In particular, models based

on partial differential equations (PDEs) that have been used for predicting tumor evolution (see for instance [19]) may be enriched and adapted to consider more complex and coupled phenomena.

One example of particular interest is modeling of the so-called called *go or grow* paradigm [20], characteristic of many tumoral processes, among which GBM is a particular case. Indeed, hypoxia has been proposed as one of the main driving forces of GBM progression [21]. The fast proliferation of GBM cells close to brain blood vessels may provoke their occlusion, which in turns leads to local hypoxia. Thus, many cells die forming a necrotic core around the collapsed vessel. The surviving cells migrate towards other non collapsed vessels, looking for oxygenated areas. These migrating structures, formed of high-density areas of cells, are called pseudopalisades [22]. Once they reach a new vessel, their migration stops and the proliferation returns. This cyclic process of migration and proliferation is known in the scientific literature as the *go or grow* paradigm. It proposes that cells exhibit a migratory or proliferative activity depending on the oxygen concentration [20]. Hypoxia Inducible Factors (HIFs) are considered as the main biomolecular activators of this activity [23–25]. Recently, we have been able to reproduce these histological structures *in vitro* [26, 27]. Also, we developed a mathematical model incorporating the *go or grow* hypothesis, which allowed us to reproduce the GBM evolution under different experimental configurations also *in vitro* [28], and to derive, from cell culture images, information on the cell behavior [29].

However, parametric models are corseted by the mathematical relations that describe the *a priori* assumed biological hypotheses, so they present an obvious modeling bias. Besides, in our experiments, we try to understand the intrinsic mechanisms that control these biological processes; a knowledge that goes further than the numerical value of a specific parameter and is generally related to concepts with a clear biological meaning, such as whether the metabolic change is sharp or smooth, localized or distributed, or presents one or many different levels of transition.

A new promising family of neural networks, named as Physically-Guided Neural Networks with Internal Variables (PGNNIV), has just emerged as a tool to identify, evaluate and derive constitutive models from observable measurements [30, 31]. The fundamental idea is to incorporate the physical knowledge on the system into the network and to concentrate the learning power of Artificial Neural Networks in the intrinsic physical mechanisms that are intended to be found up. Very recently, a similar approach combining neural networks and physical equations (Physics-Informed Neural Networks (PINNs) [32]) has been proposed as a way to discover hidden mechanistic relationships using the Fisher–Kolmogorov–Petrovsky–Piskunov equation [33] as a benchmark problem, a concept that has been coined as Biologically-Informed Neural Networks (BINNs). All the same, PGNNIV offer greater flexibility in the definition of the internal variables of interest, including the non-measurable ones. Thus they are able to unravel more complex metabolic mechanisms, such as non-local or global models [31]. Additionally, they may deal with problems involving changeful external stimuli, that is, different source terms and boundary conditions, something that PINNs cannot afford.

In this work, we demonstrate how PGNNIV allow unravelling the mathematical structure that identifies the intrinsic metabolic mechanisms associated with cell changes due to the variation of some measurable fields, as the oxygen concentration around the cell that can be measured in microfluidic devices. This identification of the detailed *go or grow* mechanism related to hypoxia allows to accurately predict the cell evolution under highly variable external stimuli, including normoxic, gradient and hypoxic configurations, without the requirement of any initially prescribed parametric relation. The data used for testing the methodology are generated *in silico*, thus allowing evaluating the predictive and explanatory capacity of PGNNIV. Indeed,

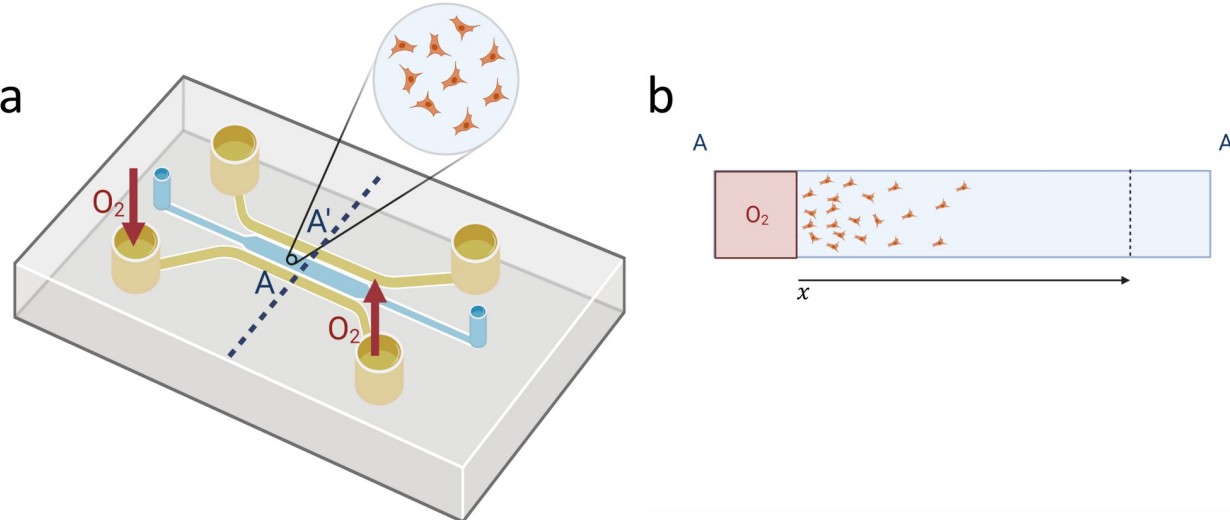

**Fig 1. Example of experimental configuration for modeling cell cultures.** Due to the long length of the lateral channels with respect to the width of the chamber, the geometry of the model is assumed as one-dimensional being the length the width of the chamber, $l$. The cell and the oxygen concentrations are associated with continuum fields $u_1 = u_1(x, t)$ and $u_2 = u_2(x, t)$ respectively. At the location of the lateral channels, $x = 0, l$, boundary conditions for both fields have to be supplied; we have represented a gradient configuration, but it is possible to feed oxygen by the two lateral channels in a symmetric configuration. (**a**) Scheme of the experimental configuration. (**b**) One-dimensional approximation of the cell culture. Created with BioRender.com.

we demonstrate that this methodology does not only permit the identification of complex metabolic changes but also improves the prediction accuracy of parametric models.

The paper is structured as follows. The materials and methods section describes the formulation here presented. First, the mathematical model for Glioblastoma evolution is briefly revised, emphasising how the role of hypoxia is commonly taken into account for modeling the *go or grow* paradigm. Then, the model under the PGNNIV framework is presented, detailing the data generation and the network training process. In the Results section, the main results of the paper are presented: the unravelling of the metabolic behavior and the ability to predict the cell evolution under different oxygenation conditions. Finally, in the Discussion section, the present and future of this methodology is discussed, while in the Conclusions, the main results and conclusions are summarised.

## Materials and methods

Our typical experimental configuration for the tests here analysed is shown in Fig 1. Here, the geometry of the model (assumed as one-dimensional, as the length of the lateral channels is much larger than the width of the chamber), and the different field variables are represented. Provided that the lengthscale is large enough, it is possible to identify the cell concentration with a continuum field $u_1 = u_1(x, t)$. Besides, the oxygen concentration is associated with a field $u_2 = u_2(x, t)$. The oxygen is supplied to the cell culture via the lateral channels. In response to this stimulus, cells will undergo migration and/or proliferation along the width of the chamber of length $l$.

### Mathematical model of Glioblastoma cell culture evolution

**Governing equations.**  Our starting point is a nonlinear reaction-diffusion system of partial differential equations that governs the evolution of GBM cells and the concentration of

oxygen in a microfluidic device [28]. Although some works support the use of two [34–36] or even three [27] phenotypes for describing the cell population, we use a previously validated model offering some flexibility for modeling the switch between proliferative and migratory activity. In particular, the equations of the fields evolution are:

$$\frac{\partial u_1}{\partial t} = \frac{\partial}{\partial x}\left(D_1 \frac{\partial u_1}{\partial x} - \chi\Pi_{\mathrm{go}} u_1 \frac{\partial u_2}{\partial x}\right) + \alpha_1\Pi_{\mathrm{gr}} u_1\left(1 - \frac{u_1}{c_{\mathrm{s}}}\right), \tag{1a}$$

$$\frac{\partial u_2}{\partial t} = \frac{\partial}{\partial x}\left(D_2 \frac{\partial u_2}{\partial x}\right) - \alpha_2\left(\frac{u_2}{u_2 + k_{\mathrm{m}}}\right)u_1. \tag{1b}$$

The first term of the R. H. S. of Eq (1a) represents the flow term associated with cell culture migration and has two contributions: the non-oriented motility term $D_1 \frac{\partial u_1}{\partial x}$ (modelled here as a random diffusion process) and the chemotaxis term $-\chi\Pi_{\mathrm{go}} u_1 \frac{\partial u_2}{\partial x}$, where cell motility is induced by the oxygen gradient. $\Pi_{\mathrm{go}}$ is a correction factor that will be discussed later. The second term corresponds to the reaction term and is associated with logistic growth [37], except for the correction term $\Pi_{\mathrm{gr}}$ that will be also explained later.

With respect to the oxygen evolution equation, Eq (1b), the first term of the R. H. S. is again the flow term, consisting solely of oxygen diffusion, and the second corresponds to the oxygen consumption by cells. The correction between brackets in the second term is the Michaelis-Menten kinetic model [38] and accounts for the kinetics of oxidative phosphorylation that occurs in the membrane of cellular mitochondria [39].

Eqs (1a) and (1b) must be complemented with appropriate boundary and initial conditions. The initial condition is a known cell profile that is seeded in the microfluidic device at the beginning of the experiment (normally constant in the whole chamber). Note that we will refer to this time as $t = 0$ even if it is not necessarily the experiment starting time, identifying the instant when the cell culture profile is measured and the microfluidic device is fully oxygenated:

$$u_1(x, t = 0) = c(x), \tag{2a}$$

$$u_2(x, t = 0) = \mathrm{O}_2^*(x), \tag{2b}$$

with $c(x)$ a given known function and $\mathrm{O}_2^*(x)$ the ambient oxygen level. The cell culture is subjected to a fixed oxygen concentration at the lateral channels. Besides, we assume that the walls of the culture chamber at the microfluidic devices are impermeable to cells, so only oxygen flow is allowed through them. In that case, the boundary conditions are:

$$\frac{\partial f_1}{\partial x}\Big|_{x=0} = 0, \tag{3a}$$

$$\frac{\partial f_1}{\partial x}\Big|_{x=l} = 0, \tag{3b}$$

$$u_2(x = 0, t) = \mathrm{O}_2^{\mathrm{L}}(t), \tag{3c}$$

$$u_2(x = l, t) = \mathrm{O}_2^{\mathrm{R}}(t), \tag{3d}$$

where we have defined $f_1 = D_1 \frac{\partial u_1}{\partial x} - \chi\Pi_{\mathrm{go}} u_1 \frac{\partial u_2}{\partial x}$ as the cell flow, $l$ is the length of the culture

chamber and $O_2^L(t)$ and $O_2^R(t)$ are known functions defining the oxygen levels at the two lateral channels aside the chamber.

At this point, the presented framework has seven model parameters, $D_1, D_2, \chi, \alpha_1, \alpha_2, c_s$ and $k_m$. Some of them have a well identified value in the scientific literature. For example:

- The oxygen diffusion, $D_2 = 1 \times 10^{-5}$ cm$^2 \cdot$ s$^{-1}$ has been reported in many works [40, 41].

- The Michaelis-Menten constant, $k_m = 2.5$ mmHg, is very particular of the specific kinetics of the reaction in hands [41, 42].

All other parameters can be easily determined in specific well-controlled experiments. For example:

- The parameters related to the logistic cell growth, $\alpha_1$ and $c_s$, can be determined in cell growth experiments under fully oxygenated conditions and in absence of oxygen gradient, both in microfluidic devices [43, 44] or using cell spheroids [45].

- The oxygen consumption rate, $\alpha_2$ is easily obtained by measuring the oxygen pressure at the ambient in an isolated system with a controlled cell culture population and for high oxygenation levels, such that the Michaelis-Menten correction, between brackets in Eq (1b), may be considered as 1. It is even possible to determine both $k_m$ and $\alpha_2$ from the oxygen pressure using an Eadie–Hofstee diagram [46] or a Lineweaver–Burk plot [47].

- The non-oxygen-mediated pedesis constant, $D_1$, is more complicated to determine as spatial cell cultures are necessary. However, spheroid cultures [48] and microfluidic devices [26] offer a great opportunity for cell migration evaluation. If full oxygenation is guaranteed in the whole culture, no oxygen gradient is formed so non-oxygen mediated pedesis is easily computed, for instance, once given $\alpha_1$, $D_1$ can be determined by evaluating the cell migration radial velocity $V$ and using the Fisher's model [49], $V = 2\sqrt{D_1 \alpha_1}$.

- The value of $\chi$ is substantially more difficult to determine. Indeed, as the cell migration depends on the oxygen level (and not only on the oxygen gradient), it is difficult to estimate this value from one single experiment or measurement. However, we can measure the cell culture migration under an oxygen gradient in a very localized region where the oxygen level may be considered almost constant [50]. Nevertheless, as it will be discussed later, we are rather interested in the overall value $\chi \Pi_{go}$. In this relation, $\chi$ is a reference value and $\Pi_{go}$ is a correction term incorporating the effect of hypoxia in the migration.

Fig 2 illustrates some schematic experiments that can be implemented to determine the model parameters appearing in Eq (1) using conventional cell culture techniques and microfluidic devices.

In addition to the model parameters, the evolution of the GBM cell culture is also influenced by the boundary and initial conditions, in terms of the functions $c(x)$, $O_2^*$, $x \in [0;l]$, $O_2^L(t)$ and $O_2^R(t)$, $t \in \mathbb{R}^+$, which play the role of problem data. That is, for the problem to be perfectly defined we need to specify the functions $c$, $O_2^L$ and $O_2^R$ together with the ambient oxygen pressure $O_2^*$.

***Go or grow* activation functions.** The metabolic behavior of the GBM cells, in particular, its response to changes in the oxygen pressure, is mathematically encoded in the functional form of $\Pi_{go}$ and $\Pi_{gr}$. These two functions regulate the activation/deactivation of both processes: migration and proliferation. There is sound evidence in the scientific literature that the switch between the proliferative and migratory activity in a cell population is hypoxia-mediated [23, 24], that is $\Pi_{go} = \Pi_{go}(u_2)$ and $\Pi_{gr} = \Pi_{gr}(u_2)$. However, there is not much knowledge about the details of this metabolic change (e.g.: are migration and proliferation simultaneous

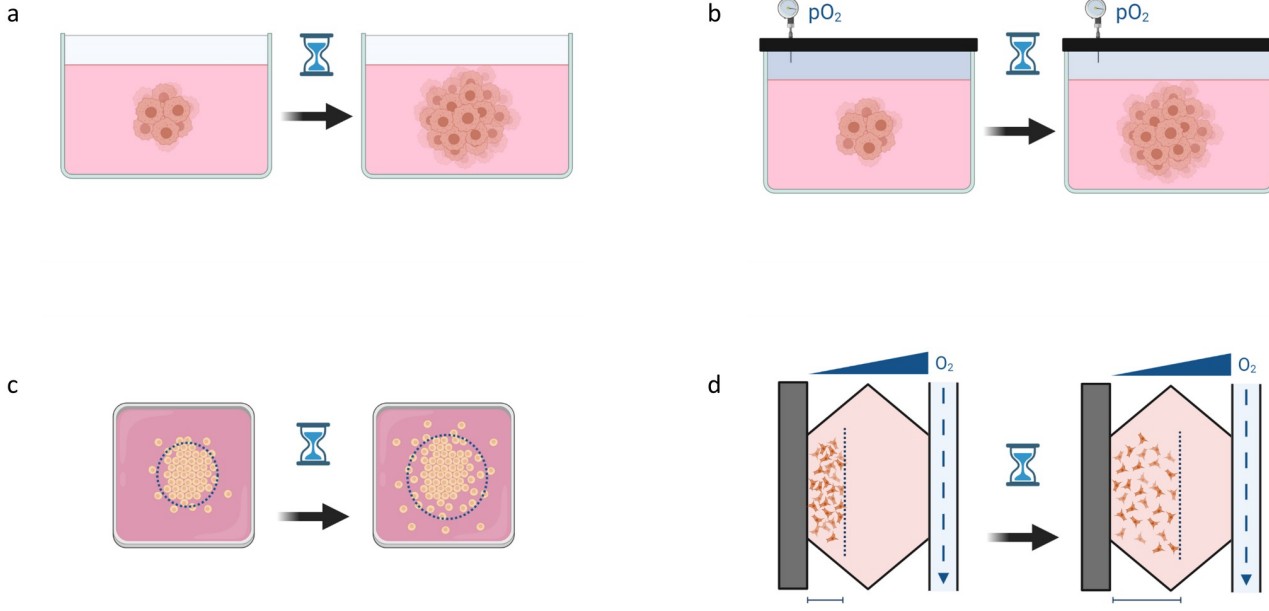

**Fig 2. Scheme of the different experiments that can be performed to obtain the model parameters.** Obtaining the source term parameters does not require a spatial cell distribution, although this is necessary for characterising the cell parameters associated with migration. (**a**) Determining $\alpha_1$ and $c_s$. (**b**) Determining $\alpha_2$ and $k_m$. (**c**) Determining $D_1$. (**d**) Determining $\chi$. Created with BioRender.com.

or not? Is the transition between them smooth? Is it strictly monotonic? Is it restricted to a narrow interval in the oxygen concentration region?).

In a recent paper [28], this transition was modelled by using piecewise linear functions of the ReLU kind, that is:

$$\Pi_{\text{go}}(u_2) = \begin{cases} 1 & \text{if} \quad u_2 \leq 0 \\ 1 - \dfrac{u_2}{\theta_{\text{go}}} & \text{if} \quad 0 < u_2 \leq \theta_{\text{go}}, \\ 0 & \text{if} \quad \theta_{\text{go}} < u_2 \end{cases} \tag{4}$$

and

$$\Pi_{\text{gr}}(u_2) = \begin{cases} 0 & \text{if} \quad u_2 \leq 0 \\ \dfrac{u_2}{\theta_{\text{gr}}} & \text{if} \quad 0 < u_2 \leq \theta_{\text{gr}}. \\ 1 & \text{if} \quad \theta_{\text{gr}} < u_2 \end{cases} \tag{5}$$

Here, $\theta_{\text{go}}$ and $\theta_{\text{gr}}$ play the role of oxygen thresholds. Additionally, it has been assumed that $\theta_{\text{go}} = \theta_{\text{gr}}$, so this model implicitly assumes that $\Pi_{\text{go}}(u_2) + \Pi_{\text{gr}}(u_2) = 1$, even if this consideration should, in principle, be modified to rely on a deeper understanding of the cell metabolism and in particular of the cell energy consumption. Indeed, the only biological evidence is that $\Pi_{\text{gr}}$ is a nondecreasing function and $\Pi_{\text{go}}$ is a nonincreasing one. The relation between the two may be, in general, more complex, assumed as unknown, or even be unveiled also by the network.

The parameter values associated with Eqs (4) and (5) provided reasonably accurate results in the characterisation of certain cell cultures. In particular, GBM culture evolution of the cell line U251-MG in microfluidic devices has been well described, even for different experimental configurations using these expressions [28]. Similar results were obtained now using Machine Learning tools (in particular, using Convolutional Neural Networks), also revealing some limitations of the parametric model [29].

However, the *go or grow* model may differ from one GBM cell line to another. Besides, the model should be adapted for other tumor families or different frameworks. Therefore, since the functional relation $u_2 \mapsto (\Pi_{go}, \Pi_{gr})$ encodes the cell metabolic changes in response to changes in the oxygen stimulus, its accurate characterisation is crucial for a complete understanding of the evolution of cell cultures, as it describes the changes that take place in the cell population behavior and consequently in the tumor progression [27, 51]. If $\mathbf{\Pi} = (\Pi_{go}, \Pi_{gr})$, the *go or grow* relation, may be written as:

$$\mathbf{\Pi} = \mathbf{\Pi}(u_2), \tag{6}$$

where $\mathbf{\Pi} : \mathbb{R}^+ \to \mathbb{R}^2$ is the unknown functional relation to be learned. Unravelling the one input-two output relation $\mathbf{\Pi}$ is therefore a key aspect in an *in silico* model able to capture tumor progression in an oxygenated medium. However, one main problem arises: as $\Pi_{go}$ and $\Pi_{gr}$ are mathematical artefacts that only make sense when considered in Eq (1), they are non-measurable variables, so there is no experimental set up that permits to measure them directly. Furthermore, the measurement of the oxygen pressure in cell culture is usually difficult due to technical considerations, even if possible under some particular conditions [52, 53]. This adds an extra difficulty when defining or calibrating the model $\mathbf{\Pi}$.

## Physically-guided neural network with internal variables

**Concept of PGNNIV.** Physically-Guided Neural Networks with Internal Variables (PGNNIV) are a generalisation of the former concept of Physics Informed Neural Networks (PINN) [32]. In this latter, the physics of the problem *informs* the network via the output variables: the physical equations constrain the values of the output variables to belong to a certain physical manifold. For instance, to ensure that they satisfy a given partial differential equation. In other words, the loss function is directly defined in terms of the problem physics. PGNNIV go one step further, as in this case, the physical equations constrain the values reached by an arbitrary number of neurons in some intermediate layers. As a consequence, it is possible to interpret some hidden features and some relationships between internal variables (IV) of the problem that now acquire a physical interpretation [30, 31]. The physics does not constrain, but only *guides* the network learning capacity, as the measured data may be supplied to endow the network with explanatory capacity.

Going into the details, a PGNNIV is a problem formulated in the following archetypal way. Let us consider a PDE system of equations that is split into:

$$\begin{aligned} \mathcal{F}(u, v) &= f, \quad \text{in } \Omega, \\ \mathcal{G}(u) &= g, \quad \text{in } \partial\Omega, \\ \mathcal{H}(u) &= v, \quad \text{in } \Omega, \end{aligned} \tag{7}$$

where $u$ and $v$ are the unknown fields of the problem, $\mathcal{F}$ and $\mathcal{H}$ are functionals representing the known and unknown physical equations of the problem in hands. $\mathcal{G}$ is a functional that specifies the boundary conditions, and $f$ and $g$ are known fields. Once discretised, Eq (7) has an analogous representation in finite-dimensional spaces in terms of vectorial functions $\mathbf{F}$, $\mathbf{G}$

and $H$ and nodal values $u$, $v$, $f$ and $g$. The Physically-Guided problem is therefore formulated as:

$$
\begin{aligned}
y &= \mathsf{Y}(x); \quad v = \mathsf{H}(u), \\
\text{s. t.} \quad x &= I(u, f, g), \\
y &= O(u, f, g), \\
R(u, v, f, g) &= 0,
\end{aligned}
\tag{8}
$$

where:

- $R$ are the physical constraints, related to the relationships given by $F$ and $G$.

- $I$ and $O$ are functions specifying the input and the output of the problem, that is, the data used as starting point to make predictions and the data that we want to predict.

- $\mathsf{Y}$ and $\mathsf{H}$ are models. $\mathsf{Y}$ is the *predictive model*, whose aim is to infer accurate values for the output variables for a certain input set and $\mathsf{H}$ is the *explanatory model*, whose objective is to unravel the hidden physics of the relation $u \mapsto v$.

A PGNNIV is built when the problem (8) is formulated in the language of Neural Networks, with an appropriate structure and topology for $\mathsf{Y}$ and $\mathsf{H}$.

## Discretised model

**Spatial discretisation.** Let us first discretise the Eq (1). This may be done by using Finite Differences (FD) or Finite Elements (FE) as it is usual when working with Partial Differential Equations (PDEs) [54]. The one-dimensional character and simple geometry of the cell culture in microfluidic devices under oxygen gradients [26] allow us to use FD to discretise the governing equations. Then, we define the nodal values of the fields $u_1$ and $u_2$ using the vectors $u_1$ and $u_2$, that is, $u_{ij} = u_i(x_j)$ where $x_j = j\Delta x$, $j = 0, \ldots, n$, is the spatial coordinate associated with a given discretisation of the domain $[0; l]$, $\Delta x = \frac{l}{n}$. The spatial derivatives may be computed using any finite difference scheme, resulting in a linear operator $D$. Then, Eq (1) results in:

$$
\dot{u}_1 = D\left(D_1 Du_1 - \chi \boldsymbol{\Pi}_{\mathrm{go}} \odot u_1 \odot Du_2\right) + \alpha_1 \boldsymbol{\Pi}_{\mathrm{gr}} \odot u_1 \odot \left(1 - \frac{u_1}{c_{\mathrm{s}}}\right),
\tag{9a}
$$

$$
\dot{u}_2 = D(D_2 Du_2) - \alpha_2(u_2 \oslash (u_2 + k_{\mathrm{m}})) \odot u_1.
\tag{9b}
$$

We have used the symbols $\odot$ and $\oslash$ for indicating pointwise multiplication and division respectively. It is important to note that $\boldsymbol{\Pi}_{\mathrm{go}}$ and $\boldsymbol{\Pi}_{\mathrm{gr}}$ are here vector functions. The framework considered permits considering functional relationships, that is, the value of $\boldsymbol{\Pi}$ at a point $x$ could depend on the value of the field $u_2$ at the whole computational domain. However, the underlying nature of the *go or grow* framework allows us to consider $\Pi(x) = \Pi(u_2(x))$, $x \in [0; l]$, or equivalently, for the vector $\boldsymbol{\Pi}$, $\Pi_j = \Pi_j(u_{2j})$, permitting to work with sparse graphs for the network topology (that is, sparse tensors and operators). In addition to the model $\boldsymbol{\Pi}$, it would be possible to consider the rest of the specific model parameters ($D_1$, $\chi$, $\alpha_1$, $c_{\mathrm{s}}$, $D_2$, $\alpha_2$, and $k_{\mathrm{m}}$) as parameters to be learned during the training process. However, as this is similar to conventional parametric fitting (except for the fact that we use the broad capabilities of NN hardware and software [30]), we consider them here as known, with their values detailed next when specifying data generation.

In order to adapt the problem to our notations, let us describe Eq (9) as:

$$\begin{aligned} \dot{\boldsymbol{u}} &= \boldsymbol{F}(\boldsymbol{u}, \boldsymbol{\Pi}), \\ \boldsymbol{\Pi} &= \boldsymbol{H}(\boldsymbol{u}). \end{aligned} \tag{10}$$

**Temporal discretisation.** With respect to the time integration, many options are possible. Multistep and Runge-Kutta methods [55] are one of the most efficient, although they are also computationally expensive. For our purposes, it is enough to consider a two-point scheme. Given the ODE $\dot{\boldsymbol{y}} = \boldsymbol{f}(\boldsymbol{y})$, we discretise it using the generalized mid-point rule, that is, by approximating $\boldsymbol{y}(t + \Delta t) - \boldsymbol{y}(t) \simeq \Delta t \boldsymbol{f}(\beta \boldsymbol{y}(t) + (1 - \beta)\boldsymbol{y}(t + \Delta t))$, $\beta \in [0; 1]$. This approximation leads to the discretisation:

$$\boldsymbol{y}^{n+1} = \boldsymbol{y}^n + (\Delta t)\boldsymbol{f}(\beta \boldsymbol{y}^n + (1 - \beta)\boldsymbol{y}^{n+1}). \tag{11}$$

With this notation, taking $\beta = 1$ we recover the forward Euler method and with $\beta = 0$ we recover the backward Euler approach.

Applying Eq (11) to Eq (10) we obtain:

$$\begin{aligned} \boldsymbol{u}^{n+1} &= \boldsymbol{u}^n + (\Delta t)\boldsymbol{F}(\beta \boldsymbol{u}^n + (1 - \beta)\boldsymbol{u}^{n+1}, \boldsymbol{\Pi}), \\ \boldsymbol{\Pi} &= \boldsymbol{H}(\beta \boldsymbol{u}^n + (1 - \beta)\boldsymbol{u}^{n+1}). \end{aligned} \tag{12}$$

Finally, we may define the residual $\boldsymbol{R}$, that is indeed the equation encoding the problem physics, as:

$$\boldsymbol{R}(\boldsymbol{u}^n, \boldsymbol{u}^{n+1}) = \boldsymbol{u}^{n+1} - \boldsymbol{u}^n - (\Delta t)\boldsymbol{F}(\beta \boldsymbol{u}^n + (1 - \beta)\boldsymbol{u}^{n+1}, \boldsymbol{\Pi}(\boldsymbol{u}^{n+1}, \boldsymbol{u}^n)). \tag{13}$$

The presented framework is generalisable to multistep and Runge-Kutta integrators. For instance, for the latter:

$$\boldsymbol{u}^{n+1} = \boldsymbol{u}^n + (\Delta t)\sum_{i=1}^{s} b_i \boldsymbol{k}_i, \tag{14}$$

with

$$\begin{aligned} \boldsymbol{k}_i &= \boldsymbol{F}\left(\boldsymbol{u}^n + (\Delta t)\sum_{j=1}^{s} a_{ij}\boldsymbol{k}_j, \boldsymbol{\Pi}\right) \quad i = 1, ..., s, \\ \boldsymbol{\Pi} &= \boldsymbol{H}\left(\boldsymbol{u}^n + (\Delta t)\sum_{j=1}^{s} a_{ij}\boldsymbol{k}_j\right), \end{aligned} \tag{15}$$

where $a_{ij}$, $b_i$ and $c_i$, $i = 1, \ldots, s$ are the particular coefficients of the selected numerical scheme.

In that case, the residual $\boldsymbol{R}$ may be written as:

$$\boldsymbol{R}(\boldsymbol{u}^n, \boldsymbol{u}^{n+1}) = \boldsymbol{u}^{n+1} - \boldsymbol{u}^n - (\Delta t)\sum_{i=1}^{s} b_i \boldsymbol{k}_i, \tag{16}$$

where $\boldsymbol{k}_i$ is given by Eq (15).

**PGNNIV construction.** The crucial part of building the PGNNIV is the definition of the network topology, as well as of the input and output layers. As explained when defining the biological problem, there is no way of straightforwardly measuring the variables $\Pi_{gr}$ and $\Pi_{go}$, so these will be our internal variables, $\boldsymbol{v} = \boldsymbol{\Pi}$, while $\boldsymbol{u} = (\boldsymbol{u}_1, \boldsymbol{u}_2)$ are the measurable ones, that is,

the cell and oxygen profiles at two consecutive time steps. Therefore:

$$I(\boldsymbol{u}, \boldsymbol{v}) \quad = \boldsymbol{u}^n, \tag{17a}$$

$$\boldsymbol{O}(\boldsymbol{u}, \boldsymbol{v}) \quad = \boldsymbol{u}^{n+1}. \tag{17b}$$

The reason for defining the input and output variables this way is to achieve accurate predictive capacity, besides the required explanatory capacity. Indeed, once the model has been trained, it is possible to predict the outcome, that is, the cell and oxygen profiles at time $t + \Delta t$, from the ones given at time $t$. Note that the cell profiles (the output) at time $t + \Delta t$ are obtained in real-time, as there is no need for solving any differential equation (we only need a network evaluation). The predictive and explanatory subnetworks are, therefore:

$$\boldsymbol{u}^{n+1} \quad = \mathsf{Y}(\boldsymbol{u}^n), \tag{18a}$$

$$\boldsymbol{\Pi} \quad = \mathsf{H}(\boldsymbol{u}_2). \tag{18b}$$

The PGNNIV graph and flow are illustrated in Fig 3. It is important to note that, although the explanatory subnetwork is the juxtaposition of two multilayer perceptrons, it is applied at each nodal value, so it acts as a convolutional network moving through the different collocation points in space. Note that if $\beta = 1$, as it is the case for this work, the input solely corresponds to the field values at time step $n$.

In a PGNNIV, the network loss function is the combination of a physics-associated term and a data-associated term. However, given the topology of the presented network, illustrated in Fig 3, all known physics of the problem is introduced explicitly in the network by means of the topology. Thus, the loss term is directly computed as:

$$L = \sum_{n=1}^{N_{\text{data}}} \sum_{i=1}^{2} \| \ \hat{\boldsymbol{u}}_i^{n+1}(\boldsymbol{u}^n) - \boldsymbol{u}_i^{n+1} \ \|^2. \tag{19}$$

where we have denoted by $\hat{\boldsymbol{u}}_i^{n+1}$ the predicted value of the field $\boldsymbol{u}$ by the PGNNIV. Recall that, observing the expression of the residual given by Eqs (13) and (19) may be written as:

$$L = \sum_{n=1}^{N_{\text{data}}} \| \ \boldsymbol{R}(\boldsymbol{u}^n, \boldsymbol{u}^{n+1}) \ \|^2, \tag{20}$$

where we have defined the residual as a function of the data, $\boldsymbol{R} = \boldsymbol{R}(\boldsymbol{u}^n, \boldsymbol{u}^{n+1})$.

Note that the architecture defining the PGNNIV, represented in Fig 3 and summarized in Eqs (18) and (20), representing respectively the predictive and explanatory network, does not depend on the initial and boundary conditions. This implies that the network may be trained with data coming from different experimental configurations, thus ensuring that the explanatory network $\mathsf{H}$ is properly represented in the data. Once trained, predictions can be made, for any external stimuli and initial state. This is indeed one of the main differences between BINNs and PGNNIVs.

## Data generation and training process

**Data for model validation.**   Here we describe the data that will be used to feed the network. It is important to note that here the data-set is generated synthetically for validation purposes, but in real-life applications, this data-set would be the result of experimental measurements.

a

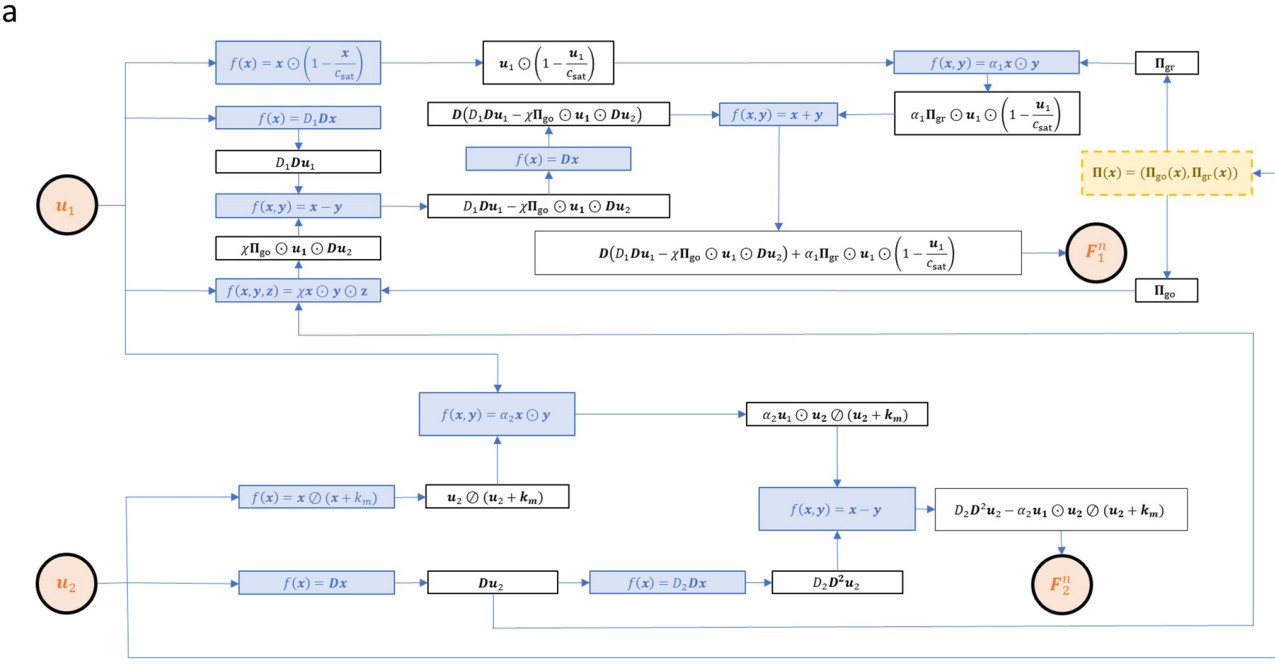

b

c

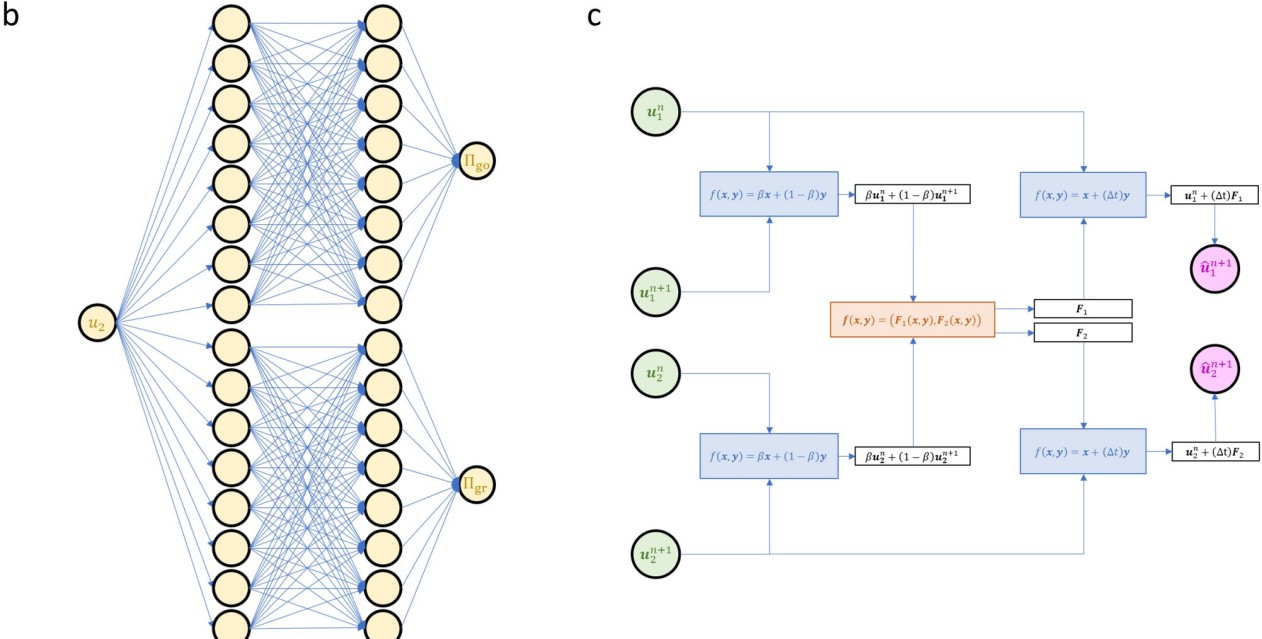

**Fig 3. Structure of the PGNNIV.** Network topology and the different operators. The measurable (independent) variables, which are treated as the input variables, are represented in green while the predicted (dependent) variables, that are treated as the output of the network, are represented in magenta. The known operators are represented in blue, the unknown operators are represented in yellow and the hybrid operators are represented in orange. For illustrative purposes, we have split the network into three subnetworks. (**a**) Components of the network related to the problem physics, Eq (9), that is the discrete representation of Eq (1). (**b**) Explanatory subnetwork, whose aim is to unravel the relationship implicit in Eq (6). (**c**) Subnetwork related to the integration procedure, that is the one relating the input and output variables.

**Table 1. Different functional relationships defined for the validation procedure.** The different functions include features such as different smoothness and nonlinearities.

| Chemotaxis activation function | Growth activation function | Parameters | Model name |
|---|---|---|---|
| $\Pi_{go}(x) = \mathbb{I}_{[0;\theta]}(x)$ | $\Pi_{gr}(x) = \mathbb{I}_{[\theta;+\infty)}(x)$ | $\theta$ | Heaviside, Binary step |
| $\Pi_{go}(x) = \left(1 - \frac{x}{\theta}\right)\mathbb{I}_{[0;\theta]}(x)$ | $\Pi_{gr}(x) = \frac{x}{\theta}\mathbb{I}_{[0;\theta]}(x)$ | $\theta$ | Piecewise linear, ReLU |
| $\Pi_{go}(x) = \frac{k}{x+k}$ | $\Pi_{gr}(x) = \frac{x}{x+k}$ | $k$ | Michaelis-Menten |
| $\Pi_{go}(x) = \frac{1}{2}\left(1 + \tanh\left(\frac{x-\theta}{\Delta\theta}\right)\right)$ | $\Pi_{gr}(x) = \frac{1}{2}\left(1 - \tanh\left(\frac{x-\theta}{\Delta\theta}\right)\right)$ | $\theta, \Delta\theta$ | Sigmoid, Logistic |

**Benchmark models.** In order to evaluate the performance of the method let us suppose four different *true* functional relationships for the metabolic model $\Pi = \Pi(u_2)$ that are described in Table 1. For illustration purposes, we assume for the different models that $\Pi_{gr}(x) = 1 - \Pi_{go}(x)$, although the PGNNIV may unravel the metabolic behavior for *true* models not satisfying this relationship.

We claim that our PGNNIV based on the governing Eq (1), encoding the known physics of the problem, is able to discover the actual biological metabolic model among the four presented in Table 1. This is possible due to the universal learning capabilities of neural networks [56, 57].

**Profile generation.** The data were generated by simulating cell profiles using Eq (1) with the boundary conditions (3). The model was first written using a dimensionless version, obtained by defining $t = T\tau$, $x = L\xi$, $u_1 = U_1 v_1$ and $u_2 = U_2 v_2$, where $T$ is a characteristic time, $L$ a characteristic length and $U_1$ and $U_2$ are characteristic cell and oxygen concentrations, obtaining:

$$\frac{\partial v_1}{\partial \tau} = \frac{\partial}{\partial \xi}\left(\bar{D}_1 \frac{\partial v_1}{\partial \xi} - \bar{\chi}\Pi_{go}v_1 \frac{\partial v_2}{\partial \xi}\right) + \bar{\alpha}_1 \Pi_{gr}v_1\left(1 - \frac{v_1}{\bar{c}_s}\right), \tag{21a}$$

$$\frac{\partial v_2}{\partial \tau} = \frac{\partial}{\partial \xi}\left(\bar{D}_2 \frac{\partial v_2}{\partial \xi}\right) - \bar{\alpha}_2\left(\frac{v_2}{v_2 + \bar{k}_m}\right)v_1, \tag{21b}$$

with boundary and initial conditions:

$$v_1(\xi, \tau = 0) = \bar{c}(x), \tag{22a}$$

$$v_2(\xi, \tau = 0) = \bar{O}_2^*(x), \tag{22b}$$

$$\left.\frac{\partial v_1}{\partial \xi}\right|_{\xi=0} = 0, \tag{23a}$$

$$\left.\frac{\partial v_1}{\partial \xi}\right|_{\xi=\bar{l}} = 0, \tag{23b}$$

$$v_2(\xi = 0, \tau) = \bar{O}_2^L(\tau), \tag{23c}$$

$$v_2(\xi = \bar{l}, \tau) = \bar{O}_2^R(\tau), \tag{23d}$$

where the dimensionless parameters and functions are:

$$
\begin{aligned}
&\bar{D}_1 = \frac{D_1 T}{L^2}, \quad \bar{\chi} = \frac{\chi T U_2}{L^2}, \qquad \bar{\alpha}_1 = \alpha_1 T, \\
&\bar{c}_s = \frac{c_s}{U_1}, \quad \bar{D}_2 = \frac{D_2 T}{L^2}, \qquad \bar{\alpha}_2 = \frac{\alpha_2 T U_1}{U_2}, \\
&\bar{k}_m = \frac{k_m}{U_2}, \quad \bar{c}(\xi) = \frac{c(x)}{U_1}, \qquad \bar{O}_2^*(\xi) = \frac{O_2^*(x)}{U_2}, \\
&\bar{O}_2^L = \frac{O_2^L}{U_2}, \quad \bar{O}_2^R = \frac{O_2^R}{U_2}, \qquad \bar{l} = \frac{l}{L}.
\end{aligned}
\tag{24}
$$

In addition to the model parameters in Eq (24), we have to consider the ones related to the different *go or grow* models described in Table 1:

$$
\bar{\theta} = \frac{\theta}{U_2}, \quad \Delta\bar{\theta} = \frac{\Delta\theta}{U_2}, \quad \bar{k} = \frac{k}{U_2}.
\tag{25}
$$

All parameters stated in Eqs (24) and (25) should have a precise biological meaning and depend on the problem physics. The values of the different parameters are reported in Table 2. Here, their value is only illustrative as they are used only for data generation, trying to make relevant all the biological phenomena.

The different cell and oxygen profiles were generated by using the method described in [58], especially suitable for parabolic partial differential equations. The system of equations was solved numerically by means of a time-space integrator based on a piecewise nonlinear Galerkin approach which is second order accurate in space, and compatible with this kind of nonlinear equations and boundary conditions, using the Matlab `pdepe` suit. Additional details may be found in [28]. A mesh size of $\Delta\xi = 1.0$ and a time step of $\Delta\tau = 0.01$ were used. As initial conditions, we set a value of $\bar{c}(\xi) = 2$ and $\bar{O}_2^*(\xi) = \bar{O}_2^L + \frac{\bar{O}_2^R - \bar{O}_2^L}{\bar{l}}\xi$. The duration of the experiment is $\tau^* = 10$. Therefore, once the temporal series of the fields (cell and oxygen profile) are generated, the output of each simulation is an array of size $[n_t, n_x, n_u]$ with $n_t = \tau^*/\Delta\tau + 1 = 1001$, $n_x = \bar{l}/\Delta\xi + 1 = 51$ and $n_u = 2$.

**Feeding the network.** In order to recreate *in silico* different Glioblastoma On-Chip experiments, we created different cell profiles by varying the boundary conditions, that is, the oxygen levels $\bar{O}_2^L$ and $\bar{O}_2^R$. Two families of configurations were simulated: symmetric and with

**Table 2. Dimensionless model parameters used for data generation.** The values are selected to make relevant all biological phenomena.

| Parameter | Value |
|---|---|
| $\bar{D}_1$ | 1 |
| $\bar{\chi}$ | 1 |
| $\bar{\alpha}_1$ | 0.5 |
| $\bar{c}_s$ | 10 |
| $\bar{D}_2$ | 1 |
| $\bar{\alpha}_2$ | 0.05 |
| $\bar{k}_m$ | 2 |
| $\bar{l}$ | 50 |
| $\bar{\theta}$ | 2 |
| $\Delta\bar{\theta}$ | 2 |
| $\bar{k}$ | 2 |

**Table 3. Experimental configurations used for data generation.** The different configurations recreate both symmetric and gradient configurations in low, medium and high oxygenated conditions (these values have to be compared with the model-associated ones, Eq (25)).

| Configuration ($k$) | $\bar{O}_2^L$ | $\bar{O}_2^R$ |
|:---:|:---:|:---:|
| 1 | 0 | 0 |
| 2 | 0 | 1 |
| 3 | 1 | 1 |
| 4 | 2 | 0 |
| 5 | 2 | 2 |
| 6 | 3 | 0 |
| 7 | 3 | 3 |
| 8 | 4 | 0 |
| 9 | 4 | 4 |
| 10 | 5 | 0 |
| 11 | 5 | 5 |

oxygen gradient. The eleven *in silico* experiments performed are reported in Table 3. Each experiment is treated, from the PGNNIV point of view, as a batch of data as illustrated in Fig 3c: the batch $k$, $k = 1, \ldots, M$, with $M = 11$, is therefore obtained by considering the $k$-th experimental configuration and the network is fed using each pair $(\boldsymbol{u}_1^n, \boldsymbol{u}_2^n)$ as input data and each pair $(\boldsymbol{u}_1^{n+1}, \boldsymbol{u}_2^{n+1})$ as output, $n = 0, \ldots, n_t - 1$. Each batch is therefore formed by an input of size $[n_t - 1, n_x, n_u]$ (from values $n = 0$ to $n = n_t - 2$) and an output of size $[n_t - 1, n_x, n_u]$ (from values $n = 1$ to $n = n_t - 1$). The objective is then learning the underlying *go or grow* model for a particular experimental condition.

**Training process.** The neural network is trained using $N = 10^3$ epochs. At each epoch, all batches associated with the experimental configurations described in Table 3 are used for the network feeding: the $M = 11$ synthetic configurations, each one of them corresponding to one batch of data (so that the network parameters are updated after each batch feed), were sampled without replacement, so they were used in a different order at each epoch iteration. $p = 80\%$ of data at each batch is used for training purposes and $1 - p = 20\%$ is used for testing the network. In total, $N \times M$ iterations of the network are considered until reaching convergence. The Adam optimizer [59] is selected with a learning rate $r = 0.001$ and an exponential decay rate of $\beta_1 = 0.8$ for the first moment and $\beta_2 = 0.8$ for the second moment are selected.

The training process takes 234 s in a core processor i7-6700 @3.4 GHz and RAM 64 GB, that is, without the use of GPU. Therefore, for an accurate comparison, we have adjusted the parametric model using also Adam optimizer, and it has taken 113 s.

## Results

As in all problems involving PGNNIV, the neural network has both predictive and explanatory capacity. To illustrate both concepts, we will discuss first the explanatory capacity of the network, which is particular to this method. For comparative purposes, we will compare the learned relationship, $\boldsymbol{H}$ in Eq (8) or $\boldsymbol{\Pi}$ in Fig 3 with standard parametric learning, where we postulate the model described by Eqs (4) and (5), also assuming $\theta_{go} = \theta_{gr}$.

Then, we will comment on the predictive capacity of the network. As this capacity is not particular to the presented method but common to all regression techniques, we will compare our results to those obtained with standard parametric fitting.

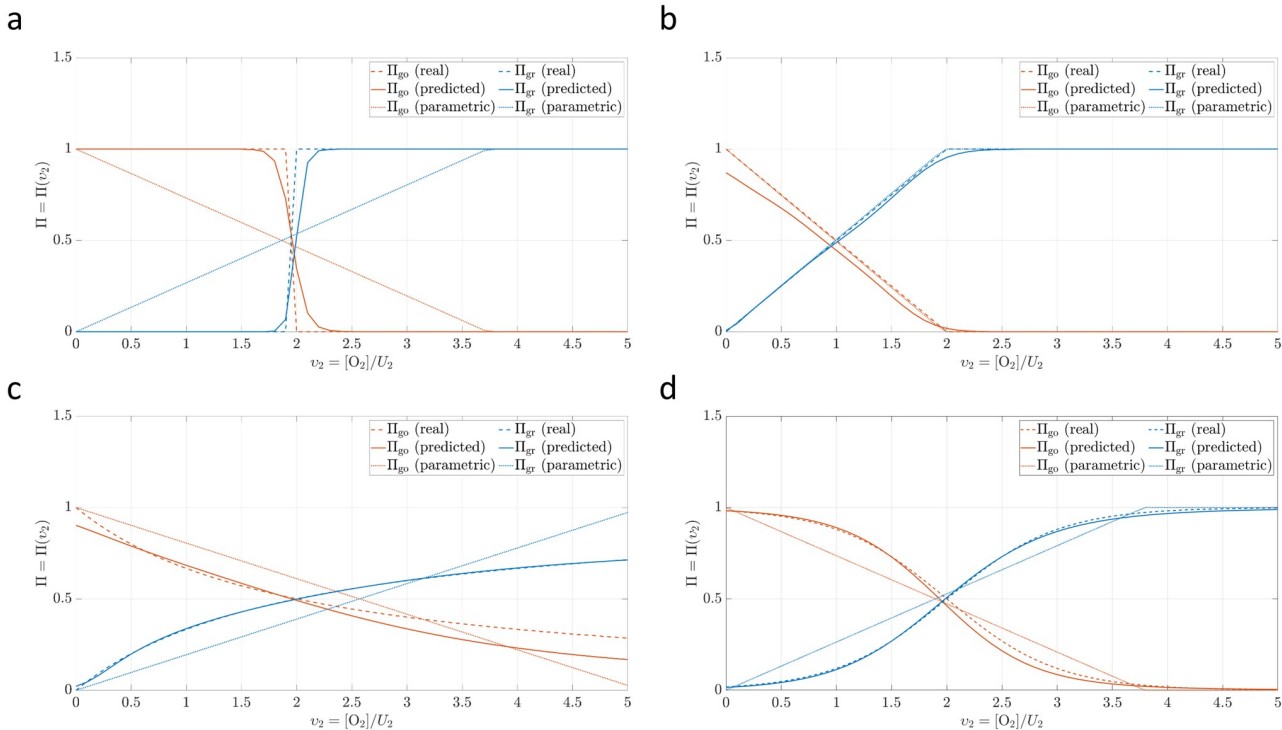

**Fig 4. Unravelling capacity of the PGNNIV.** For all models in Table 1, presenting different nonlinearities, smoothness and scales, the ground truth model is recovered correctly, especially in relation to the growth metabolic behavior. (**a**) Heaviside transition model. (**b**) ReLU transition model. (**c**) Michaelis-Menten transition model. (**d**) Logistic transition model.

## Unravelling the metabolic changes of the GBM cells

In Fig 4 we depict the learned relationship $\mathbf{\Pi}$ for the four *ground truth* models proposed in Table 1. In all cases, a good agreement is shown between the real and the predicted models. Only when the parametric family for the *go or grow* model is adequately selected, the parametric learning (that is a particular PGNNIV where the function $\mathbf{\Pi}$ is parametrized) outperforms model-free PGNNIV, as it has been reported elsewhere [30, 31]. Note that when no explicit knowledge is assumed about cell metabolism, it is difficult to either derive or postulate parametric relations such as those in Eqs (4) and (5), which are solely used as a mere instrumental tool.

Fig 5 shows the errors when unravelling the metabolic behavior $\mathbf{\Pi}$. Denoting by $\hat{\Pi}$ the model learned by the network, the error is defined as:

$$E_{\Pi}^2 = \int_0^5 \left(\hat{\Pi}(x) - \Pi(x)\right)^2 dx. \tag{26}$$

These errors are computed for both $\Pi_{go}$ and $\Pi_{gr}$. Except in the aforementioned case when the parametric family assumed includes the true model, the PGNNIV prediction clearly outperforms standard parametric approaches and keep the errors reduced for a broad family of families.

## Predicting cell culture evolution

The aim now is to explore the predictive capacity of the neural network. Once the model $\mathbf{\Pi}$ has been learned, it is represented by the multilayer perceptron topology together with all the

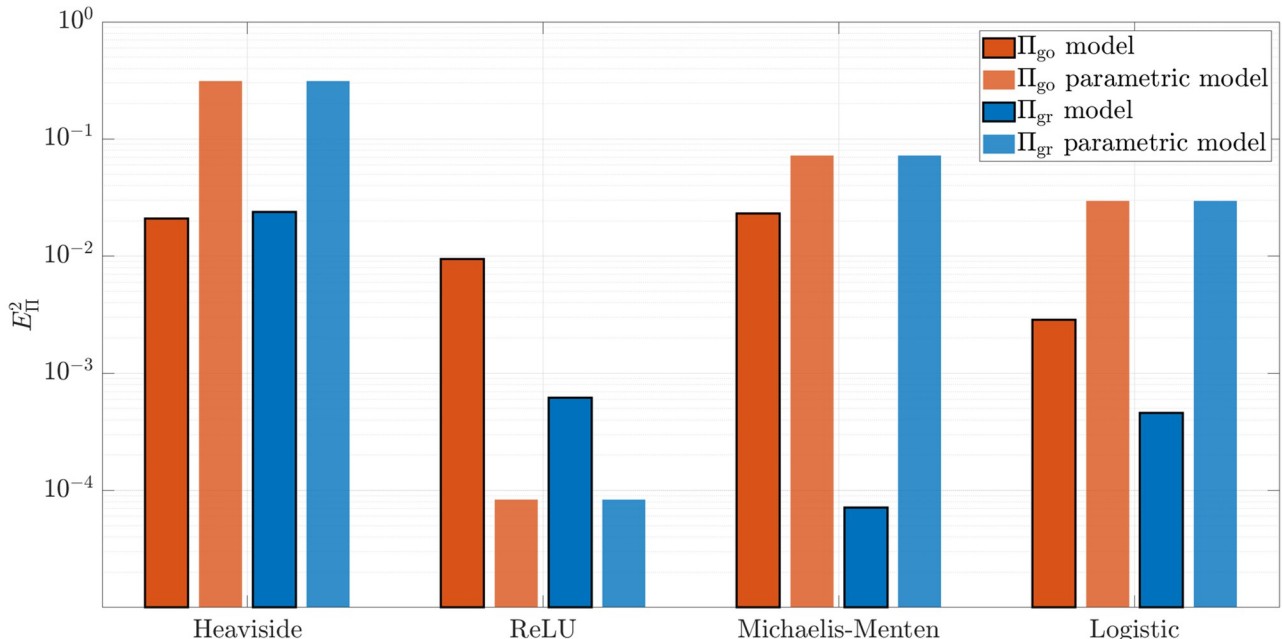

**Fig 5. Error $E_\Pi^2$ between the predicted and the real model.** The error of the prediction is robust over the different transition functions tested and outperforms any parametric fitting.

network parameters (weights and biases). Therefore it may be encapsulated as a one input—two output *black box* and inserted in any numerical integration scheme. For instance, we can consider any Runge-Kutta integrator of the form given by Eq (14) for the spatially-discretised equations, that is, to follow the approach for data generation, except for the fact that we use the learned model **Π** instead of any other of those presented in Table 1.

For illustrative purposes, let us compare the cell and oxygen profiles for three different boundary conditions: a normoxic configuration where $\bar{O}_2^L = \bar{O}_2^R = 4$, a hypoxic configuration where $\bar{O}_2^L = \bar{O}_2^R = 0$ and a gradient configuration where $\bar{O}_2^L = 0$ and $\bar{O}_2^R = 4$. Our aim is to predict the cell profile after $\tau = 20$. The results are shown in Fig (6), where we have represented for each cell profile the real one (the derived when using directly the function in Table 1), the one predicted after fitting the parametric model described by Eqs (4) and (5) and the one predicted by PGNNIV. For all the models tested, a good agreement is shown between the predicted and the real profiles, and PGNNIV always outperforms the prediction of the parametric models, except, as it was explained before, for the ReLU case. The improvement of the prediction is particularly significant for the gradient configurations (Fig 6b). Indeed, the specific features of the model have a greater impact for oxygen levels in the transition between normoxic and hypoxic behavior since it is in this case when the differences between the different models most influence the cell evolution.

In order to explore quantitatively the improvement, we define the error associated with the cell prediction as:

$$E_{\text{cell}}^2 = \int_0^{50} \left( \hat{v}_1(\xi, \tau = \tau^*) - v_1(\xi, \tau = \tau^*) \right)^2 d\xi. \tag{27}$$

In Fig 7 we compare the error of the cell prediction given by Eq (27) when estimating the cell profile using the parametric approach and the one based on PGNNIV.

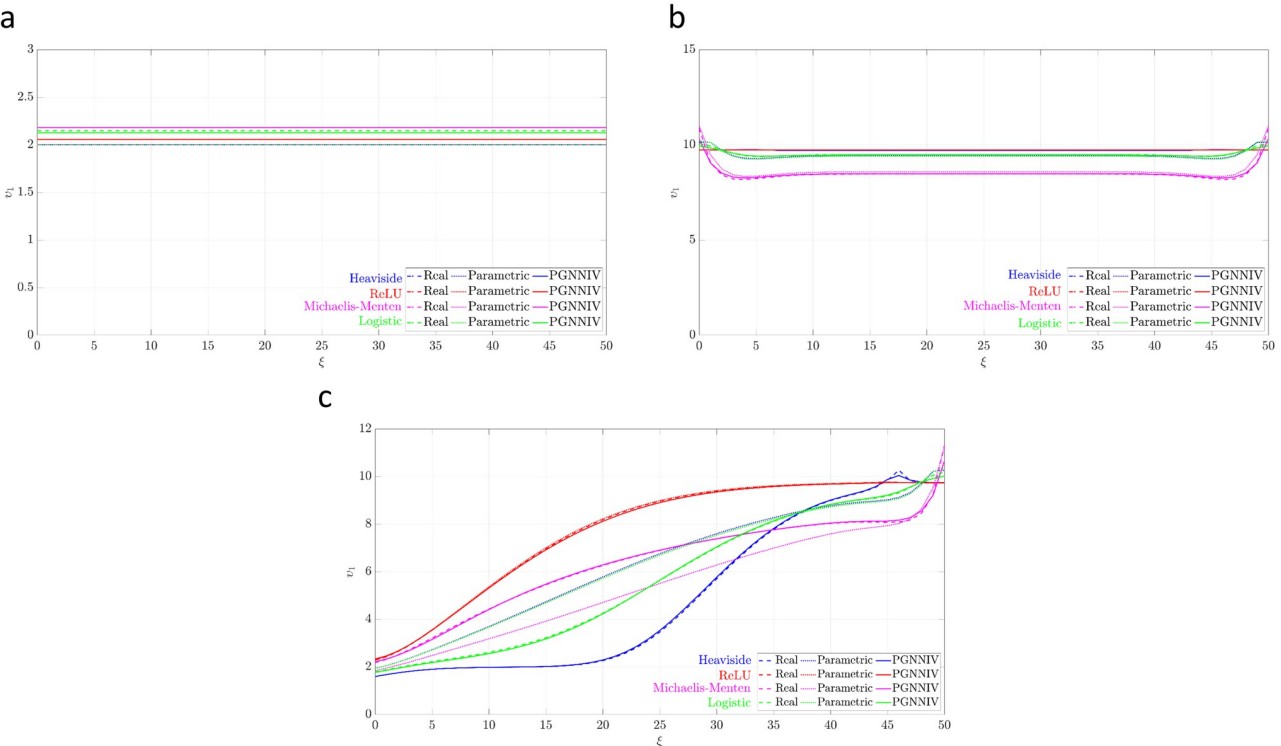

**Fig 6. Prediction of the cell profile at $\tau^* = 10$ for the different models tested and different experimental configurations.** PGNNIV improves the prediction (when compared to the parametric model) of all cell profiles, slightly for the normoxic configuration and significantly for the gradient configuration. (**a**) Hypoxic configuration. (**b**) Gradient configuration. (**c**) Normoxic configuration.

It is important to note that the training data-set was used for simulations for $\tau \le \tau^*$ with $\tau^* = 10$, so we explore here the prediction capacity of the network out of the region defined by the training data-set.

## Computational requirements

In order to evaluate the computational requirements of the method, we compare it to standard parametric fitting performance. Recall that parametric fitting depends on the algorithm selected, in terms of both time and memory requirements. Conventional algorithms, such as Levenberg-Marquardt [60] are more memory demanding and require a large number of evaluations for large datasets, when compared for instance to Stochastic Gradient Descent (SGD) with small batch sizes. Therefore, for an accurate comparison, we have adjusted the parametric model using a similar PGNNIV (using Adam optimizer with same hyperparameters) except for the fact that the model function $\mathbf{\Pi}$ was totally parametrized in terms of $\theta_{\mathrm{go}} = \theta_{\mathrm{gr}} = \theta$.

The training process for the non-parametric PGNNIV takes 234 s in a core processor i7-6700 @3.4 GHz and RAM 64 GB, compared to the 113 s that takes for the parametric PGNNIV, using TensorFlow package (Python). The differences are obviously related to the number of network parameters, that are $2 \times (8 \times 8 + 8) = 144$ for the non-parametric approach and 1 for the parametric one. Even so, in the two approaches, the low computational requirements are due to the NN framework used to face the problem.

 

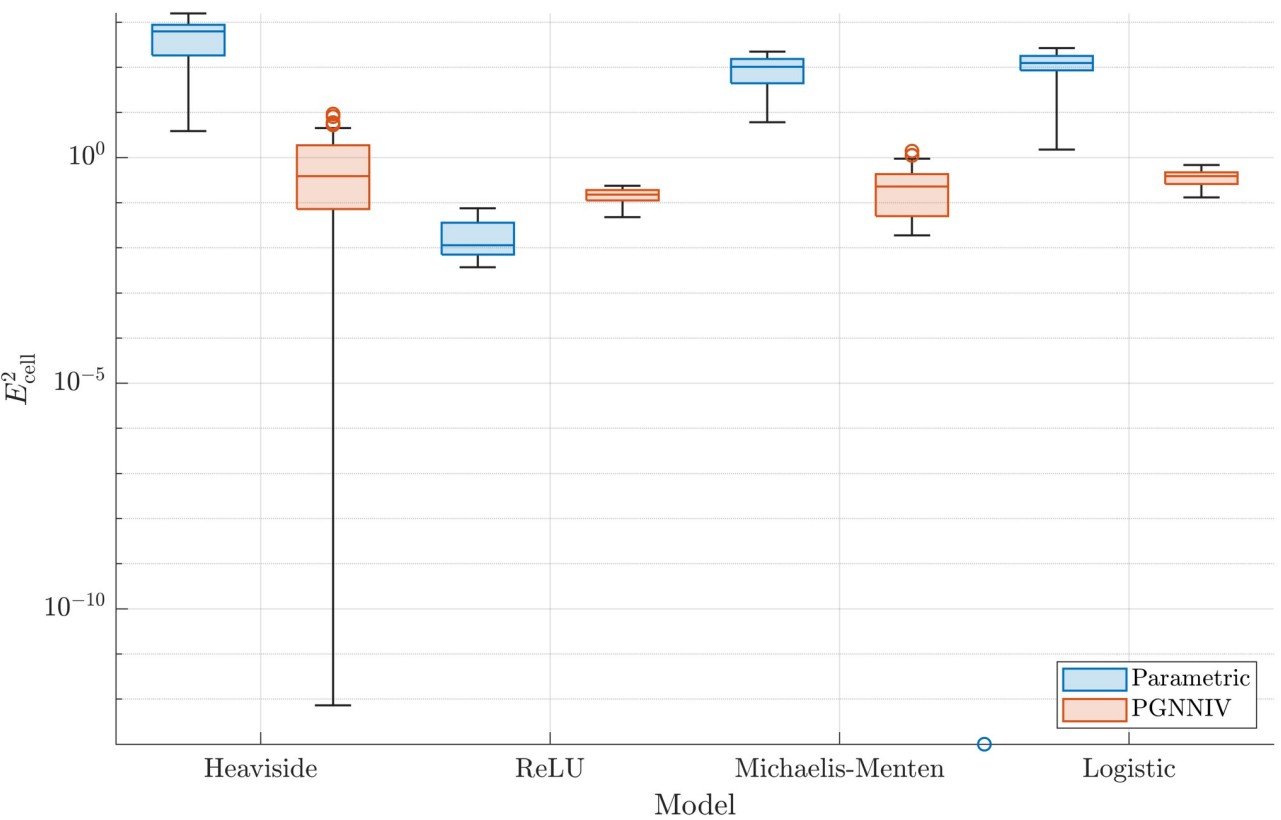

**Fig 7. Prediction error.** The non-parametric model built using the PGNNIV approach is able to correctly estimate the cell profile evolution for any arbitrary model better than specific parametric ones. The predictions are more accurate (lower error) and more precise (lower variability) along with the different tested boundary conditions.

## Discussion and open possibilities

### The present: Characterisation of complex biological cell processes

Discovery of hidden cell metabolisms is a major concern in biology. Indeed, unravelling the changes of the cell behavior when exposed to different stimuli can put us on the track of mechanisms driving the different cell signalling paths [61, 62]. As a result, parameters as the *hypoxic threshold* [28] are replaced by richer behaviors, as the ones illustrated in Fig 8.

Moreover, from a mechanistic perspective, different metabolic paths and schemes may be tested *in silico* using computational approaches [63], in order to decide whether a path candidate is compatible with the metabolic changes discovered by means of the PGNNIV. Therefore, this knowledge on the macroscopic cell metabolic behavior at the population level is important not only from an epistemic point of view, for modeling purposes, but also as a promising tool for molecular biologists, in their attempt to isolate and define the different signalling pathways, thus providing a deeper understanding on the underlying mechanisms.

Sometimes, there are some energetic constraints (the more fundamental ones are those given by the first and second principles of thermodynamics) that restrict the accessible states in a biological system [64–66]. These constraints are translated into macroscopic ones in a continuum population model. For instance, one possible constraint is the former hypothesis that $\Pi_{gr} + \Pi_{go} = 1$. However, this is a special case of the more general constraint $G(\Pi_{gr}, \Pi_{go}) = 0$, that could be founded on an energetic argument about the resources available for the cell to

**Fig 8. Parametric vs non-parametric approaches.** The degree of information about the cell metabolism is richer in the non-parametric approach.

grow or proliferate. All these extra constraints may be incorporated in the PGNNIV framework in a direct and straightforward fashion either by expressing some relational equations between variables explicitly, or by adding appropriate penalty terms in the loss function obliging to fulfil a mathematical constraint, such as $p\|G(\Pi_{gr}, \Pi_{go})\|^2$, with $p$ a penalty parameter [30].

A last remark is that PGNNIV, as any method inspired in Neural Networks, can be used as a universal approximator of the hidden interaction mechanisms between different cell populations, thus incorporating the ingredients of population sociology in systems biology [67]. For instance, if many cell populations are considered, $C_1, \ldots, C_n$, one may establish many *ad-hoc* interrelation mechanisms, $\lambda = \lambda(C_1, \ldots, C_n)$, where $\lambda$ is any model functional parameter, describing for instance migration or proliferation. The crosstalk between different cell populations has demonstrated to be important in many cellular processes such as those presented here [68, 69]. Of course, this interrelation would be properly learned if:

- The known physics and biology of the system is well enough described in terms of specific mathematical equations. That is, all known mechanisms are explicitly stated, but only them. This enables the PGNNIV to concentrate its unravelling power in the unknown interrelations.

- The available data is large enough to capture the prescribed dependency (as in the problem presented in this work). This is commonly difficult in many experimental sciences, particularly in biology, but new tools and trends such as microfluidics are promising in this regard.

As in any machine learning approach, care must be taken when interpreting the results and deriving conclusions, as the learning methods suffer from overfitting. A suitable strategy (train and test approaches, cross-validation, validation trials...) is therefore crucial for drawing generalisable conclusions.

## The future: Towards *in silico* personalised medicine

This work presents a method for going from cell expression at the tissue level, that is, the formation of cellular structures such as pseudopalisades in GBM invasion, to cell behavior at the

population level in response to the ambient stimuli. In one sense, it presents a link between clinicians and molecular biologists. A tumor biopsy may be extracted from one patient, cultivated and monitored in microfluidic devices where it may be exposed to different stimuli. The microfluidic devices have demonstrated to be capable of *in silico* reproducing histological features such as necrotic cores and pseudopalisades [26, 27], which can be captured using image and video techniques. PGNNIV is a tool able to infer, from the culture response to several stimuli, the intrinsic model of the cell reaction to such stimuli. This ability to integrate the knowledge of the response to different stimuli is a particular capability of PGNNIV, which puts them one step ahead of other methods such as BINNs in unravelling cell metabolism.

Once the intrinsic model is learned, its generalisable capability is much wider than the one offered by the histological features, as the latter is the response to very specific conditions (in mathematical terms, to very specific boundary conditions). In a sense, this strategy is the same as parametric fitting, except for the fact that there is no need for making any assumption about the model functional structure, provided that we know the internal variables we want to relate. This last issue is not minor, but is, indeed, the main objective of biology research: to make scientific conclusions about the effect and association of chemical factors with biological response.

This extra generalisation capability, when combined with appropriate mathematical models, offers new possibilities in *in silico* drug and treatment design. Once the cell response to the different stimuli is unravelled, we may variate the different stimuli levels and the different conditions. Again, in mathematical terms, this is represented by boundary conditions, initial conditions and source terms of the associated partial differential equations. Consequently, we can evaluate, again from a tissue perspective, more than from a cell population point of view, the general histological features that happen in the cell virtual tissue, which is called a Virtual Digital Twin. These histological features are responsible of key factors in cancer progression such as vessel occlusion [70], intravasation, extravasation and metastasis [70, 71], necrosis and activation of inflammatory response and/or angiogenesis [72, 73], in particular for GBM [74]. Therefore, *in silico* tests will accelerate the design of new drugs and therapies as they allow to test the hypotheses in a more flexible, faster and cheaper way.

It is important to note that all the here described sequence of steps relies on one single patient-specific biopsy, thus turning all this approach fully patient-specific, grounding this approach within the global framework of personalised *in silico* medicine [75]. From a specific patient, we infer specific histological and cellular features and, therefore, the different *in silico* trials are adapted to their particular disease, running away from parametric models, whose universality is corseted by their particular functional form.

The whole process is illustrated in Fig 9, where all the steps are summarized both in conceptual terms and using the particular example illustrated in this work. To conclude, PGNNIV is a tool for transferring the knowledge on the tissue response to knowledge on the cell gene expression. This knowledge is the one that enables to work on the cell expression level and to generalise both to arbitrary conditions and individual patients.

## Conclusions

The emerging technology of microfluidics has brought to the field of cell culture not only the possibilities of biotechnology research in more realistic biomimetic environments but also the chance of incorporating data-intensive tools such as Artificial Intelligence and Machine Learning. The term coined for this framework is Intelligent Microfluidics.

Here, we have illustrated how PGNNIV may be a valuable tool to infer the cell response at the population level from the cell response at the tumor microenvironment level in response

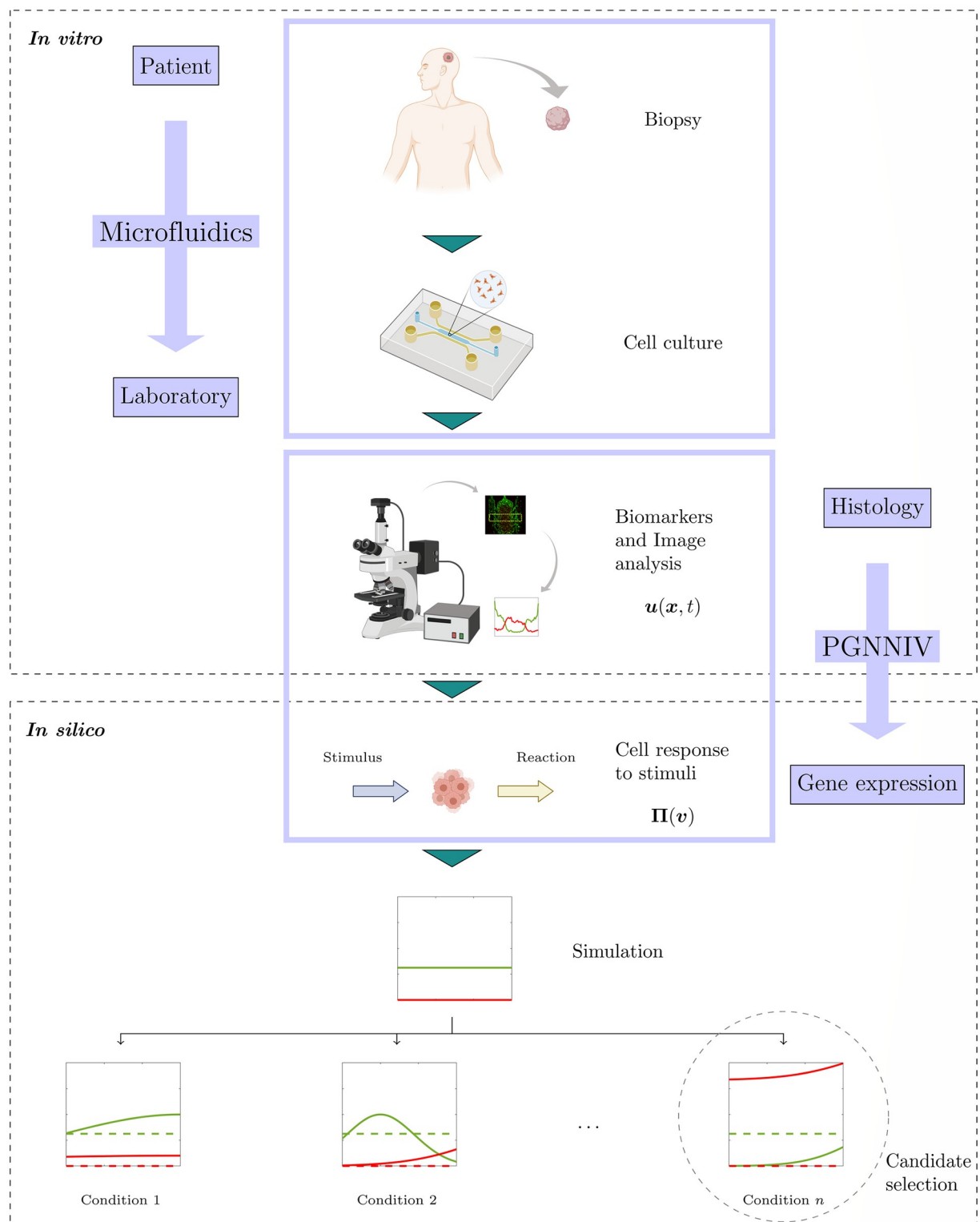

**Fig 9. Summary of the described framework.** Starting from the clinical patient, we can recreate histological features associated with the tumoral tissue in the microfluidic devices, extract the cell response to stimuli using PGNNIV and use the information to evaluate different therapies and drug candidates. Created with BioRender.com.

to external stimuli. Unlike other Physically-Informed Data Science methods, in PGNNIV the physics does not constrain, but only *guides* the network learning capacity, as the measured data may be supplied to endow the network with explanatory capacity. Using *in silico* data, we have proven the unravelling capacity of PGNNIV for different benchmark test models, thus allowing us to work in model-free and non-parametric frameworks. Indeed, PGNNIV based simulations outperform the explanatory ability of any parametric model, except, at most, if the selected model belongs to the selected parametric family, which is, in practice, a strong and unrealistic assumption.

In addition, this explanatory ability is directly translated into an improvement of the prediction for different ambient conditions. The predictive ability does not only improve the one associated with classical parametric fitting but is also independent of the underlying model, thus making unnecessary the assumption of extra hypotheses, for example about cell metabolism. This improvement is achieved regardless of the environmental conditions, even if these ones are not used during the training process. In a certain sense, the ability of PGNNIV to predict out of the range of the training dataset is exploited here at its best.

The flexibility of PGNNIV allows that any information about biological systems may be incorporated totally or partially into the computations. This includes cell-cell interactions or cell-substrate interaction among other cues. In an opposite way, it is possible to focus the learning power on any relationship between fields of interest (biological, chemical, mechanical. . .) that is intended to be learned or quantified, both for theoretical (learn about the molecular metabolic processes) or for experimental (describe the main features of the process) purposes.

The presented methodology let us glimpse some steps in the direction of personalised medicine, as it is model-free, it allows to work with tissues extracted from different patients without the need of the specification of any particular model that would ruin out the method generalisation ability. Once characterised, the tumoral population may be *in silico* subjected to different stimuli and conditions, corresponding to different exploratory treatments. The response may be analysed from a clinician point of view, that is, at the tissue level, in order to evaluate the treatment success or failure. This strategy of *in silico* test-evaluation is quick and cheap and can strongly reduce animal experimentation, therefore facilitating research in areas such as biotechnology and biomedical engineering.

## Author Contributions

**Conceptualization:** Jacobo Ayensa-Jiménez, Manuel Doblare.

**Formal analysis:** Jacobo Ayensa-Jiménez.

**Funding acquisition:** Mohamed H. Doweidar, Jose A. Sanz-Herrera, Manuel Doblare.

**Investigation:** Jacobo Ayensa-Jiménez.

**Methodology:** Jacobo Ayensa-Jiménez, Manuel Doblare.

**Project administration:** Jose A. Sanz-Herrera, Manuel Doblare.

**Resources:** Jose A. Sanz-Herrera, Manuel Doblare.

**Software:** Jacobo Ayensa-Jiménez.

**Supervision:** Mohamed H. Doweidar, Jose A. Sanz-Herrera, Manuel Doblare.

**Visualization:** Jacobo Ayensa-Jiménez.

**Writing – original draft:** Jacobo Ayensa-Jiménez.

**Writing – review & editing:** Jacobo Ayensa-Jiménez, Mohamed H. Doweidar, Jose A. Sanz-Herrera, Manuel Doblare.

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
