## [Decision Letter · Decision Letter 0]

17 Dec 2021

Dear Dr Doblare,

Thank you very much for submitting your manuscript "Understanding glioblastoma invasion using the unravelling power of physically-guided neural networks with internal variables" for consideration at PLOS Computational Biology.

As with all papers reviewed by the journal, your manuscript was reviewed by members of the editorial board and by several independent reviewers. In light of the reviews (below this email), we would like to invite the resubmission of a significantly-revised version that takes into account the reviewers' comments.

We cannot make any decision about publication until we have seen the revised manuscript and your response to the reviewers' comments. Your revised manuscript is also likely to be sent to reviewers for further evaluation.

Sincerely,

Inna Lavrik

Associate Editor

PLOS Computational Biology

Feilim Mac Gabhann

Editor-in-Chief

PLOS Computational Biology

Reviewer's Responses to Questions

**Comments to the Authors:**

Reviewer #1: The paper has a very well explain new method where it will help to modelling microfliudic techniques. The method is well explain with all the details and simulation results.

I found the paper well estructurare and written.

Reviewer #2: The review is uploaded as an attachment.

**Have the authors made all data and (if applicable) computational code underlying the findings in their manuscript fully available?**

Reviewer #1: Yes

Reviewer #2: **No: **The data and code is not available

PLOS authors have the option to publish the peer review history of their article (what does this mean?). If published, this will include your full peer review and any attached files.

Reviewer #1: **Yes: **Pilar Guerrero

Reviewer #2: No
---

## [Editor Report · Decision Letter 1]

15 Mar 2022

Dear Dr Doblare,

We are pleased to inform you that your manuscript 'Understanding glioblastoma invasion using physically-guided neural networks with internal variables' has been provisionally accepted for publication in PLOS Computational Biology.

Best regards,

Inna Lavrik

Associate Editor

PLOS Computational Biology

Feilim Mac Gabhann

Editor-in-Chief

PLOS Computational Biology

---

## [Editor Report · Acceptance letter]

30 Mar 2022

PCOMPBIOL-D-21-01502R1 

Understanding glioblastoma invasion using physically-guided neural networks with internal variables

Dear Dr Doblare,

I am pleased to inform you that your manuscript has been formally accepted for publication in PLOS Computational Biology. Your manuscript is now with our production department and you will be notified of the publication date in due course.

With kind regards,

Anita Estes
